# Predicting Transmission Suitability of Mosquito-Borne Diseases under Climate Change to Underpin Decision Making

**DOI:** 10.3390/ijerph192013656

**Published:** 2022-10-21

**Authors:** Kate Sargent, James Mollard, Sian F. Henley, Massimo A. Bollasina

**Affiliations:** 1School of GeoSciences, University of Edinburgh, Edinburgh EH9 3FE, UK; 2School of Social Science, University of Dundee, Dundee DD1 4HN, UK

**Keywords:** malaria, dengue fever, Zika virus, climate change, mosquito-borne diseases, transmission suitability

## Abstract

The risk of the mosquito-borne diseases malaria, dengue fever and Zika virus is expected to shift both temporally and spatially under climate change. As climate change projections continue to improve, our ability to predict these shifts is also enhanced. This paper predicts transmission suitability for these mosquito-borne diseases, which are three of the most significant, using the most up-to-date climate change projections. Using a mechanistic methodology, areas that are newly suitable and those where people are most at risk of transmission under the best- and worst-case climate change scenarios have been identified. The results show that although transmission suitability is expected to decrease overall for malaria, some areas will become newly suitable, putting naïve populations at risk. In contrast, transmission suitability for dengue fever and Zika virus is expected to increase both in duration and geographical extent. Although transmission suitability is expected to increase in temperate zones for a few months of the year, suitability remains focused in the tropics. The highest transmission suitability in tropical regions is likely to exacerbate the intense existing vulnerability of these populations, especially children, to the multiple consequences of climate change, and their severe lack of resources and agency to cope with these impacts and pressures. As these changes in transmission suitability are amplified under the worst-case climate change scenario, this paper makes the case in support of enhanced and more urgent efforts to mitigate climate change than has been achieved to date. By presenting consistent data on the climate-driven spread of multiple mosquito-borne diseases, our work provides more holistic information to underpin prevention and control planning and decision making at national and regional levels.

## 1. Introduction

There are multiple vector-borne diseases that impact people’s lives globally, the majority of which are spread via mosquitoes [1]. Approximately 17% of all infectious diseases are vector-borne and currently more than half the global population is at risk, the most vulnerable being children in the world’s poorest communities where living conditions are poor, immunity is low and malnourishment is common [2]. Although death is the most serious outcome, these diseases can also lead to permanent disability, pain, disfigurement, stigma, social exclusion, loss of earnings and high medical bills, ultimately impeding economic development and leaving children even more vulnerable.

Children are particularly at risk of malaria, such that an estimated two-thirds of the approximately 409,000 deaths from malaria in 2019 were children under the age of 5 [2]. Dengue fever is the most important mosquito-borne viral disease in the world [3], and although it rarely results in death, severe dengue fever can be fatal, with the largest proportion of deaths among children. Zika virus is usually not fatal; however, it can lead to congenital malformations when contracted during pregnancy and increases the risk of neurologic complications in both adults and children [4].

Although control, prevention and treatment are available for these diseases, monitoring risk is essential to support decision-making for prioritisation and allocation of resources. This is an aim of projects such as the Malaria Atlas Project [5], which explores new and innovative techniques to map current malaria distributions for prevention control and planning. However, as climate change proceeds, the distributions of these diseases are also expected to change.

Previous studies have shown that climate change is likely to cause shifts in spatial and temporal exposure to these diseases and that the severity of these impacts will depend on the particular future scenario. For example, Messina et al. [6] showed that dengue risk in 2050 is likely to contract from regions of Africa and India, and that areas in the Americas, Australia and East Asia are likely to become newly suitable. These changes will be most severe under the ‘business as usual’ emissions scenario, with decreases in risk under stabilised emissions scenarios. Ryan et al. [7] showed that there will be ‘substantially increased’ areas of suitability for Zika virus in North America and Europe by 2050 under the worst-case climate change scenario, but that limiting emissions could reduce the expansion of this suitability. Ryan et al. [8] showed shifting seasonality of malaria across the African continent, with reductions in transmission suitability in Western Africa and increases in Eastern and Southern Africa by 2050, which are most severe under the worst-case future scenario. These studies provide a valuable picture of the changing risk of mosquito-borne diseases under climate change; however, there is no research of this kind to date that uses the most up-to-date climate projections from the Coupled Model Intercomparison Project Phase 6 (CMIP6), nor any that focuses on transmission suitability for multiple diseases combined, using a unified methodology. This study aims to fill these knowledge gaps.

Studies modelling the distribution of mosquito-borne diseases employ two main approaches. The statistical (top-down) approach starts with empirical data and applies complex statistical methods incorporating a wide range of covariates that impact upon disease transmission (e.g., [6]). The mechanistic (bottom-up) approach models the biological processes that determine habitat suitability for mosquitoes and parasites (e.g., [7]). Statistical models have been shown to align largely with mechanistic outputs for the present day [9] and are often more accurate due to accounting for external influences such as human intervention and control measures such as insecticide-treated nets or health services. However, statistical methods are limited in their ability to isolate the effects of the multiple interacting covariates of environment, populations and behaviours, and by the assumption that current processes will remain constant in an unprecedented future [10]. Although statistical methods have proven to be more accurate in predicting current disease distribution at smaller scales, mechanistic modelling is widely recognised as being more suitable for large geographic areas and future predictions under climate change [11,12,13].

This study employs a unified mechanistic model to estimate the number of months of transmission suitability for malaria, Zika virus, dengue fever and all three diseases combined at the global scale in the present day and by 2050 under two climate change scenarios from the most up-to-date climate model projections. The two future shared socio-economic pathway (SSP) scenarios selected, SSP1-2.6 and SSP5-8.5, span the range of plausible global warming projections, as used in the recent Intergovernmental Panel on Climate Change Sixth Assessment Report (IPCC AR6) [14]. Using the most recent projections from the CMIP6 is important for an up-to-date, holistic view of the impact of the potential differences between climate change trajectories on vulnerable populations and in order to support climate change mitigation strategies, as well as disease prevention and control. The single metric for the three major mosquito-borne diseases combined facilitates more efficient and effective future decision-making, planning and reporting for the wide spectrum of stakeholders across a range of environmental and geo-political contexts, for example UNICEF’s Children’s Climate Risk Index [15].

The principal conclusions are that dengue fever and Zika virus are expected to undergo both geographical and temporal increases in transmission suitability, which will be more significant under the highest-emissions global warming scenario. Malaria shows a different pattern, whereby both climate change scenarios result in an overall temporal and geographical decrease of transmission suitability. However, both scenarios will also lead to significant increases in the duration of transmission suitability for concentrated areas of Central Africa. Populations in some of these areas have not previously been exposed to malaria and therefore are lacking in immunity. These results indicate that some tropical locations will become suitable for year-round transmission of all three diseases, increasing the burden of climate change on populations that are already highly vulnerable. This paper strengthens the case for intensifying and accelerating efforts to mitigate climate change, and for raising awareness on the climate-driven spread of multiple mosquito-borne diseases to support efficient future planning and decision making for disease prevention and control.

## 2. Materials and Methods

This study adopts a simplified and consistent mechanistic methodology for each disease, providing the potential of expansion to include other mosquito-borne diseases that are sensitive to changes in temperature.

### 2.1. Identifying Areas with Suitable Conditions for Transmission

The model identifies areas with suitable conditions for transmission of the mosquito-borne diseases. It is worth noting that the model does not identify where the disease is present. The model is based on a methodology developed by Ryan et al. [7,8,16,17] for examining the impact of changing temperatures under climate change to predict the future spread of different mosquito-borne diseases. These methodologies use a single basic cut-off for the thermal interval in which viral transmission is possible [16], based on a sophisticated method that links the basic reproduction number R_0_ for mosquito-borne viruses to temperature derived from laboratory experiments [10].

### 2.2. Selection of Thermal Limits

Thermal limits based on mosquito and virus traits that drive transmission of the malaria pathogen *Plasmodium*
*falciparum* (*Anopheles gambiae and other related*
*Anopheles* and *Plasmodium species*); Zika virus (*Aedes aegypti*) and dengue fever (*Aedes aegypti and Aedes albopictus*) were selected to reflect potential transmission suitability of these diseases (Table 1; see previous studies by Ryan et al. [7,8,16] for more detail on the selection of the thermal limits). As there are two thermal limits for dengue fever (one for each *Aedes* vector), a certain location and month was deemed suitable for dengue transmission if conditions were suitable for either vector.

### 2.3. Climate Data and Projections

The thermal limits were applied to gridded monthly mean temperature data representing the present day and 2050 under two climate change scenarios, to enable comparison across time and space as well as the potential differences in transmission suitability under different climate trajectories. Gridded monthly mean temperature data from 2001-2020 at 1 degree resolution were obtained from the Berkeley Earth Surface Temperatures dataset [18] to obtain a 20-year mean, to account for the interannual variability in temperatures.

The two climate pathways chosen were shared socioeconomic pathway (SSP) 1-2.6 and SSP5-8.5 to allow comparison of the best- and worst-case climate trajectories. SSP1-2.6 is the more optimistic sustainability pathway based on international policy agreements and emissions reductions, which aims to limit average global temperature rises to 1.5 °C above pre-industrial levels. SSP5-8.5 is the ‘business as usual’ pathway, with continued fossil-fuel development [19]. Monthly temperature data were taken from 10 models from the Scenario Model Intercomparision Project (ScenarioMIP), part of the CMIP6. This is a subset of models, selected to ensure availability of data in both scenarios, and a sufficiently high horizontal resolution that they could be interpolated to a 1 × 1 degree common grid to match the present-day temperature observations. Temperature projections were derived by adding the difference in mean monthly temperatures between the period 2040-2060 and the final 15 years of the modelled historical simulation (2000-2015) to the BEST present day climatological temperatures.

### 2.4. Aridity Mask

Mosquitoes require a moist environment for breeding, and this is a factor in determining transmission suitability of the diseases studied. Due to the limitations of using precipitation as an indicator of habitat suitability (see [8]), the normalised difference vegetation index (NDVI) has been widely used as an environmental covariate proxy for the water component of the habitats of medically important arthropods [20,21,22]. In this study, an aridity mask was applied to exclude areas that are too dry for mosquito lifecycle completion. The mask was calculated using remotely-sensed NDVI, with locations that are ‘too dry’ defined as two consecutive months where the NDVI falls below a threshold of 0.125 [23]. Gridded monthly average NDVI for 2020 at 1 degree resolution was taken from NASA Earth Observations’ Moderate Resolution Imaging Spectroradiometer (MODIS) on the Terra satellite [24]. As future projections of NDVI are not currently available, NDVI values for 2020 were held constant for the 2050 projections.

### 2.5. Evaluating Transmission Suitability

The number of months of transmission suitability for each disease at a specific location was defined as the aggregate of the number of months where the average monthly mean temperature falls within the thermal limits and the NDVI for two consecutive months does not fall below 0.125. Following [8], we defined transmission suitability of 10–12 months as endemic, 7–9 months as seasonal, 4–6 months as moderate and 1–3 months as marginal.

To obtain an overall number of months of transmission suitability for all the diseases combined, a mean of the number of months of transmission suitability for each disease was taken. Each disease was weighted equally as there has been no evidence to date to support ranking of the severity of globally important mosquito-borne diseases.

Previous studies [7,8,16] have used gridded 5 arc-minute (approximately 10 km^2^ at the equator) monthly mean temperature data from the Worldclim dataset [25] for the baseline, and CMIP5 data from between four and six Global Circulation Models (GCMs) for Representative Concentration Pathways (RCPs) 4.5 and 8.5. This study allows an analysis using the most trusted [26] present-day temperature data [27] and up-to-date climate projections, enabling a comparison of the best- and worst-case scenarios, that incorporates the socioeconomic narratives of the SSPs. Furthermore, previous studies have only included an aridity mask for modelling malaria [8]. This study applies the aridity mask for all the diseases, for consistency, and because all mosquitoes rely on the presence of moisture to breed and complete their lifecycle.

### 2.6. Statistical Analysis

Statistical significance of the differences between future projections and present-day data, and the differences between the two future scenarios, was assessed by performing a two-tailed Student’s *t*-test. Where differences are plotted in the results figures, only those that were statistically significant at the 95% confidence level (*p* < 0.05) were included.

## 3. Results

### 3.1. Malaria Transmission Suitability

Malaria is currently endemic in Africa, south of the Sahara, and in parts of Central and South America and South Asia [28]. This is reflected in the present-day results (Figure 1a), which show that most of the areas suitable for malaria transmission are located in the tropics, and large parts of the regions where malaria is known to be endemic are suitable for transmission for 10 months of the year or more.

Projections for 2050 show that even under the best-case climate change scenario, areas suitable for transmission of malaria will shift significantly (Figure 1b). Although a large area of Central Africa will remain suitable for endemic malaria transmission under both scenarios (Figure 1b,c), there is potential for a notable decrease in transmissibility of up to 9 months, which is most evident in the Democratic Republic of Congo (DRC) (Figure 1d,e). However, potential increases in transmissibility of up to 11 months are possible in the bordering countries to the south and east of the DRC under both climate change scenarios (Figure 1d,e). The same can be seen in South America, with endemicity persisting most significantly in southeast Brazil and the northwest of the continent, but with the duration of transmission suitability reducing to zero over much of the Amazon basin (Figure 1b–e).

Transmission suitability can be seen to expand to the south and north of the tropical regions that are typically suitable for transmission under both future scenarios (Figure 1b–e), such that many locations that were previously unsuitable are due to become suitable. New exposure outside the tropics is at worst seasonal, with less than or equal to 8 months transmission suitability under both climate change scenarios. Figure 1f shows that SSP5-8.5 will result in a larger geographic expansion in transmission suitability to the north and south of currently endemic regions, along with larger reductions in duration within the tropical region, than SSP1-2.6. In the majority of areas where the increase in transmission suitability is greater under scenario SSP5-8.5 than SSP1-2.6, the difference is small (1–2 months), although this corresponds to a larger geographic expansion reaching further north into Russia, southern and eastern Europe, the USA and areas of Australia. Differences between scenarios of 5–7 months can be seen in small areas of northern South America and southern Africa.

Areas that are not currently suitable for transmission but become suitable under SSP1-2.6 are projected to experience between 1 and 7 months of new suitability, although the majority of new suitability is for 1 or 2 months and this is largely concentrated in the northern temperate zone (Figure 1a,b,d). Under SSP5-8.5, areas of new suitability of between 1 and 3 months are much more widespread across the northern temperate zone and increase to over 6 months in some parts of the northern Andes, Central America, Central Africa and Indonesia (Figure 1a,c,e).

### 3.2. Zika Virus Transmission Suitability

The present-day results show that Zika virus has endemic transmission suitability in large areas of the tropics, including much of northern South America, parts of Central America, Sub-Saharan Africa, South and Southeast Asia and parts of northern Australia (Figure 2a). In contrast to malaria, there is a notable increase in Zika transmission suitability by 2050 and comparatively few areas where decreases are projected (Figure 2b–e). Under both projected climate change scenarios, decreases of between 1 and 3 months occur in small parts of northwest Australia, north and northwest Sub-Saharan Africa and India and Pakistan (Figure 2d,e).

Endemic areas expand across Central Africa and some northwestern parts of South America under both scenarios (Figure 2b–e). Increases in transmission suitability between the present day and 2050 are up to 9 months under SSP1-2.6 (Figure 2d), and up to 10 months under SSP5-8.5 in Central Africa (Figure 2e). Northwestern South America has up to 6 months more transmission suitability under SSP1-2.6 (Figure 2d) and up to 11 months more under SSP5-8.5 (Figure 2e), compared to present day.

The few areas that become newly suitable for transmission under SSP1-2.6 show suitability for between 1 and 2 months in temperate regions, up to 9 months in some parts of Central Africa and up to 7 months in the Northern Andes. Under SSP5-8.5, newly suitable areas are more widespread and suitability lasts longer, increasing to 3 months in temperate zones and 11 months in the tropics.

The number of months of transmission suitability show increases in duration and expansions geographically both north and south of the tropics under both climate change scenarios. Figure 2f shows that transmission suitability is largely between 1 to 3 months longer under SSP5-8.5 than SSP1-2.6 in suitable areas, with a difference of up to 5 months in northwest South America and Central Africa.

### 3.3. Dengue Fever Transmission Suitability

Figure 3a shows that transmission suitability for dengue fever is endemic across the tropics in the present day. There are also considerable areas of seasonal and moderate transmission suitability (4 to 9 months) across the whole of Australia, southern USA, southern Europe, central Asia and southeast China.

Figure 3b,d show new suitability in temperate regions of up to 1 or 2 months under SSP1-2.6, with potential for 1 month of suitability as far north as northern Russia. Similar to Zika, the highest increases in transmission suitability, up to 9 months, are in northwestern South America with up to 5 months of new suitability. Eastern Africa shows up to 6 months more suitability in some areas. Areas becoming suitable for year-round transmission are in eastern and southern Central America, central and northwestern South America, northern Australia and parts of Southeast Asia.

Under SSP5-8.5, areas becoming suitable for year-round transmission are in similar locations as SSP1-2.6 (Figure 3c,e). Outside the tropics, transmission suitability covers a greater area and duration increases by up to 3 months. Eastern Africa shows increases of up to 6 months, and in parts of northwestern South America up to 10 months. The largest decreases in suitability under both scenarios are up to 3 months in northern Sub-Saharan Africa, illustrating that the prevalence of dengue is to expand in area and duration, with minimal decreases. Conditions newly suitable for transmission occur as far north as the Arctic Circle under SSP5-8.5, although this is limited to one month of suitability.

### 3.4. Combined Disease Transmission Suitability

When all diseases are combined, to inform policy making, transmission suitability reduces in duration across the tropical regions, but remains notable in large areas of Central Africa and northern South America, under both climate change scenarios (Compare Figure 4a–e). Areas moving to year-round suitability are also concentrated in these regions under both scenarios.

Increases in transmission suitability extend beyond the typical boundaries of the tropics, with longer periods across Africa and South America. The greatest increase across all diseases is in the northern Andes; up to 4 months under SSP1-2.6 (Figure 4d) and almost 6 months under SSP5-8.5 (Figure 4e). Although this region shows an overall reduction in areas suitable for malaria transmission, it will remain endemic with dengue fever and increase from no risk of Zika virus to 7 months under SSP1-2.6 or 11 months under SSP5-8.5. Other areas at greatest risk from all diseases are in Central Africa (despite the decline in malaria in some areas), which could see increases in transmission suitability of between 3 and 4 months under both climate change scenarios (Figure 4d,e).

Shorter periods of transmission suitability are projected to spread into previously unaffected areas in Europe, North America and Russia. In the northern temperate zone under both climate change scenarios these new areas are suitable for between 1 and 2 months (Figure 4d,e), whilst SSP5-8.5 indicates larger and more northerly areas being affected (Figure 4f). Although the differences in transmission suitability between the best- and worst-case climate change scenarios are small for all diseases combined, between 2 and 3 months, Figure 4f shows that considerably more areas across the globe could see the introduction of these diseases under SSP5-8.5.

## 4. Discussion

### 4.1. Transmission Suitability Risk and Implications

The 2050 projections of transmission suitability for malaria indicate considerable decreases in Central Africa, as well as new areas of year-round transmission suitability mainly in the south and east of Central Africa and the northern Andes under both climate change scenarios. Some of these areas that have not previously been suitable will become so for up to 9 months of the year under SSP5-8.5, posing a significant risk to populations that lack existing immunity. The results suggest that although total malaria transmission suitability will decrease more under SSP5-8.5 than SSP1-2.6, the geographical reach will extend further for shorter periods of transmission and areas that remain suitable will experience longer periods of transmission suitability under SSP5-8.5. These findings align with previous research related to Africa [8]. However, the more up-to-date data used in this study indicate that these shifts could happen more quickly than previously estimated and even under the best-case climate change scenario (SSP1-2.6).

Dengue fever is currently the most spatially and temporally prevalent disease investigated in this study in terms of transmission suitability. This prevalence in both time and space will increase under both climate change scenarios. Zika virus follows the same pattern, but to a lesser degree. The results for dengue and Zika align broadly with previous studies using the same methodology [7,16]. In particular, our results reinforce the most recent results from Ryan et al. [16]; namely, that year-round transmission suitability will continue to be focused in the tropics, with suitability declining with increasing distance from the equator and expanding into temperate regions for 1 to 2 months of the year under both scenarios. Ryan et al. [16] also showed that under higher emissions scenarios, expansion of transmission suitability increases geographically and temporally for *A. aegypti* mosquitoes, but not *A. albopictus*, as temperatures become too warm for the latter. For this study, conditions are considered suitable for transmission of dengue fever if they are suitable for either *Aedes* vector. Whilst this differs from Ryan et al. [16], the pattern of transmission suitability under SSP5-8.5 shown here aligns with those results, although some areas are excluded due to the inclusion of the aridity mask in this study. Ryan et al. [7] indicate a net increase in Zika virus transmission suitability, as very few regions become too hot for transmission for at least one month of the year. This leads to an increase in year-round Zika virus transmission suitability in tropical regions, and many areas in temperate zones becoming newly suitable for at least one month of the year. However, as specified in the previous study [7], the geographic expansion in transmission suitability for Zika virus is more restricted than that of dengue fever, which has a lower thermal bound, allowing transmission of dengue in cooler regions. This study aligns with these previous results, although the aridity mask employed here for all diseases excludes areas such as the Sahara from transmission suitability where conditions are too dry for mosquito reproductive success.

Zika virus and dengue fever currently have much larger geographic coverage of endemic transmission suitability than malaria, and this will increase further as global warming intensifies, particularly under the worst-case scenario. In contrast to malaria, dengue fever and Zika virus are predicted to experience much fewer decreases in both temporal and geographic extent of transmission suitability under both climate change scenarios. This does not mean that malaria should be considered a retreating challenge as the consequences remain significant, particularly for children, who are at greater risk of death from malaria both than adults and when compared to their risk of dying from dengue or Zika. However, considering these three major diseases together does illustrate potential for reallocation of resources for prevention and control of malaria to newly and more severely affected regions and towards prioritising the prevention and control of Zika virus and dengue fever across the worst affected tropical regions.

The widespread risk areas for dengue fever and Zika virus make prioritising interventions difficult, but it is essential to ensure suitable actions to safeguard global health now and into the future. Interventions that are currently widely used include insecticides, which can have detrimental environmental consequences and lead to increasing mosquito resistance; removal of mosquito breeding sites; information campaigns to increase public awareness of how to avoid mosquito bites; use of insecticide-treated nets; and surveillance, testing, research and development to collect and analyse data to increase understanding and forecasting of these diseases and their consequences. This kind of research includes understanding the severity of the consequences of exposure to different or multiple diseases, such as the effect on previously unexposed populations who are more prone to serious outbreaks (e.g., [29]), or understanding the factors that affect the severity of symptoms and put people at risk. This is especially relevant for children who are the most vulnerable to these diseases and their consequences, such as congenital Zika syndrome and other birth defects.

Emerging technologies for controlling these diseases include: vaccines, although these are not widely used and there is no vaccine to date for the Zika virus; the sterile insect technique, which introduces large numbers of bred sterile male mosquitoes to reduce reproduction; and the introduction to male mosquitoes of the *Wolbachia* bacteria, which reduces the numbers of *A. aegypti* mosquitoes and has been shown to be effective against the transmission of Zika virus [30] and dengue fever [31]. These strategies must be coordinated by suitable responses from global health systems, including the World Health Organisation, in order to focus political attention at national and international levels [29].

Global public health systems have been criticised for failure to act decisively in response to health emergencies and inequities of resource allocation [29,32]. Critics of the response to the COVID-19 pandemic recommend a ‘multipronged approach that targets threats collectively’ and ensures a global, equitable response, such that all countries can respond to new threats [32]. Learning from responses to a range of health emergencies can help to ensure preparedness, and the use of predictive data can add to that preparation to identify where resources will be required and plan an effective strategy. Furthermore, a rapid response, adequate resources and communication to improve public confidence have also been identified as key factors for successful response strategies to health emergencies [32]. Data to estimate the potential future spread of known diseases, such as those presented here, will help to focus the allocation of resources where they are needed most as the risk of future unknown diseases, epidemics and pandemics increases.

An editorial shared by over 200 global health journals highlights the link between climate change, socio-economic inequities and pandemics, and the urgent need to act on climate change, reduce inequity and support global coordination to avoid irreversible, catastrophic harm to health and severe implications for all countries and communities [33]. The authors also reiterate that high-income countries must do more to support low- and middle-income countries in mitigation and adaptation efforts, including improving the resilience of health systems, building local capacity and capabilities, empowering communities and educating the public about the health risks of the climate crisis. Making data about the projected future spread of diseases publicly available, such as that available for the current distribution of malaria through the Malaria Atlas Project [5], can not only improve planning and decision making, but also increase public awareness and political will.

Climate-related health burdens are expected to be much higher without further investment in mitigation and adaptation strategies, especially for low-income countries who are more vulnerable to mosquito-borne diseases, are the least responsible for greenhouse gas emissions to date and have the fewest resources to adapt to climate change and its myriad impacts [34]. This strengthens the case for the urgent intensification of efforts to mitigate climate change globally and to minimise the ever-increasing costs of adaptations required now and into the future, which will include significant prevention and control measures for an increasing global disease burden. The increasing northerly spread of Zika and dengue, particularly under the worst-case climate change scenario, must also act as a warning to the global north that it too could become increasingly susceptible to mosquito-borne diseases. This will increase the burden on public health resources, which are already strained as a result of the COVID-19 pandemic.

Research connecting climate change and adverse health outcomes has been criticised for being fragmented [34]. Our results, by considering and combining multiple mosquito-borne diseases, are useful for providing a more complete and societally relevant picture. We identify the regions most impacted by multiple diseases and how these impacts vary under different scenarios, to underpin decision-making around climate change. Our results reinforce the importance of following the best-case climate change scenario (SSP1-2.6) in order to minimise the impacts associated with mosquito-borne diseases. As such, our work strongly supports the case for intensifying climate change mitigation measures at the global scale and limiting global temperature increases to within 1.5 °C above preindustrial levels, in accordance with the Paris Agreement (2015). The data will be useful for planning and decision making on where and how to allocate resources, identifying resource requirements and potentially enhancing efficiencies by implementing interventions that can reduce risk to multiple diseases at once. Furthermore, in combination with data on other climate change risks and vulnerabilities, our results can help to identify the populations, particularly children (e.g., [15]), that are most at risk from climate change impacts to aid decision-making for planning and intervention across the full range of stakeholders.

### 4.2. Limitations and Recommendations for Future Study

We acknowledge the uncertainty inherent in climate projections and that there are a number of other variables that could have an impact on disease reproduction and transmission that are not included in our study. These potential drivers include urbanisation and population growth [6], and the impact of extreme weather events, such as heavy rain which could wash away breeding sites, but are very difficult to predict [8]. Furthermore, the study does not include aridity projections under climate change scenarios. As it is likely there will be climate-induced changes in aridity by 2050, this will affect the ability of mosquitoes to reproduce. Inclusion of aridity projections under climate change could improve the model predictions in future. The results are also limited to 1 × 1 degree resolution, due to lack of availability of up-to-date climate projections at a higher resolution. Use of higher-resolution climate projections, when available, would improve the versatility and utility of the model at smaller scales.

The results are also limited by the use of thermal limits that were obtained in a laboratory environment and therefore do not reflect the daily fluctuations of temperature that occur in the real world. This limited the application of thermal limits to daily average temperature. Further study of the thermal limits for disease transmission suitability under fluctuating daily temperatures would allow for modelling using the daily minimum and maximum temperatures, allowing for a better understanding of disease transmission under real-world conditions. It is also likely that mosquitoes and parasites will have some capacity to adapt to the changing environmental conditions in future beyond the current limitations applied to transmission [16], which is not accounted for in this study. For example, higher temperatures could accelerate the parasite or the mosquito lifecycle, as has been seen with chikungunya virus [35].

Finally, the method for combining the diseases is based on equal weighting for all three mosquito-borne diseases; further research is required to weight the impacts of these diseases according to risk. Many of these limitations also apply to other mechanistic models for predicting transmission suitability; however, this study has provided a unified methodology for estimating the impact of temperature changes under climate change on three of the most important mosquito-borne diseases, which could be expanded to more diseases where the thermal limits of transmission are known.

## 5. Conclusions

The results of this study show that by 2050 there will be notable increases in periods of transmission suitability of mosquito-borne diseases across the globe under climate change scenarios SSP1-2.6 and SSP5-8.5, with the largest increases projected under the worst-case SSP5-8.5 scenario. We also show a smaller occurrence of decreases in transmission suitability in areas that are currently suitable for endemic transmission over the same period. Our results strengthen the case for global efforts to limit climate warming, particularly to reduce the impact on tropical regions, where in the majority of cases populations are already highly vulnerable to climate change, are the least responsible for it and have the fewest economic resources to respond to its effects. However, we also show that the higher-income countries that are considered less likely to suffer the more serious effects of climate change will become increasingly susceptible to mosquito-borne diseases as temperatures rise.

The results highlight that many areas that are currently suitable for endemic malaria transmission will see a decline in transmission suitability. As such, the focus for future prevention and control of mosquito-borne diseases will need to increase the emphasis on Zika virus and especially dengue fever, both of which are projected to experience substantial increases in transmission suitability in time and space.

As these diseases have particularly serious consequences for children, organisations and policy makers concerned with children and young people are encouraged to use the results of this study to identify populations of children that live in areas that are suitable for mosquito-borne disease transmission now and in the future. This should also be combined with estimates of other climate change risks to determine what the most important risk factors are for children living in countries across the globe and where best to focus policies and interventions to safeguard and enhance the resilience of children and their communities to the impacts of climate change.

## Figures and Tables

**Figure 1 ijerph-19-13656-f001:**
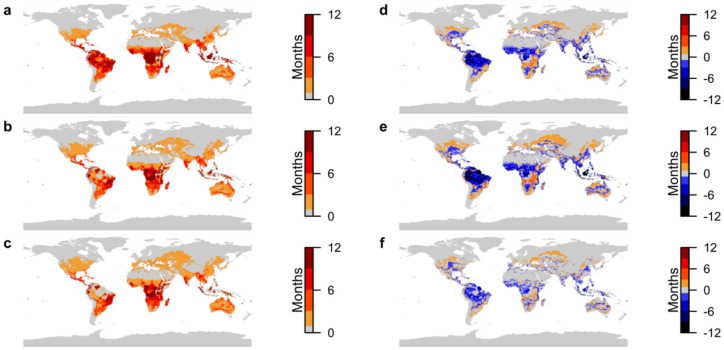
Projections of transmission suitability for malaria. (**a**) Number of months of transmission suitability in present day; (**b**) number of months of transmission suitability in 2050 under SSP1-2.6; (**c**) number of months of transmission suitability in 2050 under SSP5-8.5; (**d**) change in number of months of transmission suitability from present day to 2050 under SSP1-2.6; (**e**) change in number of months of transmission suitability from present day to 2050 under SSP5-8.5; (**f**) difference in number of months of transmission suitability in 2050 between SSP5-8.5 and SSP1-2.6. For (**a**–**c**), colour-bar scales are set so that increasing intensity of red depicts areas where transmission suitability is marginal (1–3 months), moderate (4–6 months), seasonal (7–9 months) and endemic (10–12 months). For (**d**–**f**), colour-bar scales are similarly grouped, with increasing intensity of red depicting increases in transmission suitability of 1–3 months, 4–6 months, 7–9 months and 10–12 months, and increasing intensity of blue depicting the equivalent decreases. Only areas where differences are statistically significant at the 95% confidence level are plotted.

**Figure 2 ijerph-19-13656-f002:**
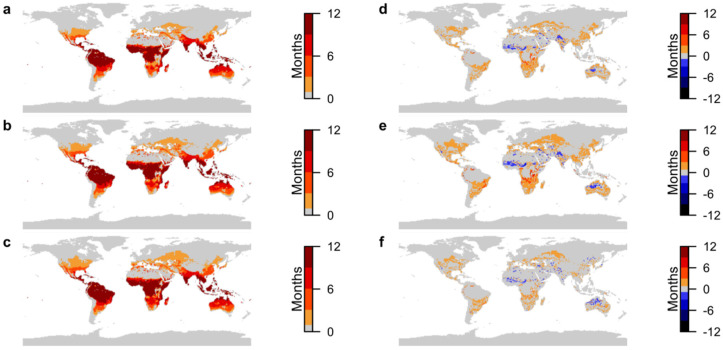
Projections of transmission suitability for Zika virus. (**a**) Number of months of transmission suitability in present day; (**b**) number of months of transmission suitability in 2050 under SSP1-2.6; (**c**) number of months of transmission suitability in 2050 under SSP5-8.5; (**d**) change in number of months of transmission suitability from present day to 2050 under SSP1-2.6; (**e**) change in number of months of transmission suitability from present day to 2050 under SSP5-8.5; (**f**) difference in number of months of transmission suitability in 2050 between SSP5-8.5 and SSP1-2.6. For (**a**–**c**), colour-bar scales are set so that increasing intensity of red depicts areas where transmission suitability is marginal (1–3 months), moderate (4–6 months), seasonal (7–9 months) and endemic (10–12 months). For (**d**–**f**), colour-bar scales are similarly grouped, with increasing intensity of red depicting increases in transmission suitability of 1–3 months, 4–6 months, 7–9 months and 10–12 months, and increasing intensity of blue depicting the equivalent decreases. Only areas where differences are statistically significant at the 95% confidence level are plotted.

**Figure 3 ijerph-19-13656-f003:**
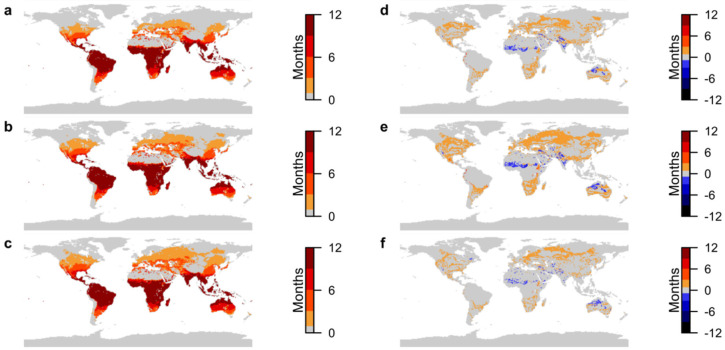
Projections of transmission suitability for Dengue fever. (**a**) Number of months of transmission suitability in present day; (**b**) number of months of transmission suitability in 2050 under SSP1-2.6; (**c**) number of months of transmission suitability in 2050 under SSP5-8.5; (**d**) change in number of months of transmission suitability from present day to 2050 under SSP1-2.6; (**e**) change in number of months of transmission suitability from present day to 2050 under SSP5-8.5; (**f**) difference in number of months of transmission suitability in 2050 between SSP5-8.5 and SSP1-2.6. For (**a**–**c**), colour-bar scales are set so that increasing intensity of red depicts areas where transmission suitability is marginal (1–3 months), moderate (4–6 months), seasonal (7–9 months) and endemic (10–12 months). For (**d**–**f**), colour-bar scales are similarly grouped, with increasing intensity of red depicting increases in transmission suitability of 1–3 months, 4–6 months, 7–9 months and 10–12 months, and increasing intensity of blue depicting the equivalent decreases. Only areas where differences are statistically significant at the 95% confidence level are plotted.

**Figure 4 ijerph-19-13656-f004:**
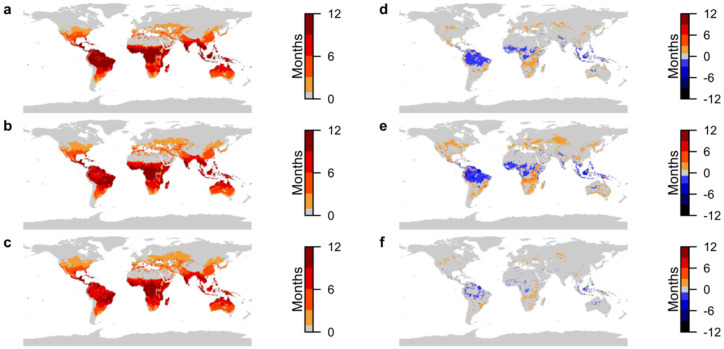
Projections of transmission suitability for all diseases combined. (**a**) Number of months of transmission suitability in present day; (**b**) number of months of transmission suitability in 2050 under SSP1-2.6; (**c**) number of months of transmission suitability in 2050 under SSP5-8.5; (**d**) change in number of months of transmission suitability from present day to 2050 under SSP1-2.6; (**e**) change in number of months of transmission suitability from present day to 2050 under SSP5-8.5; (**f**) difference in number of months of transmission suitability in 2050 between SSP5-8.5 and SSP1-2.6. For (**a**–**c**), colour-bar scales are set so that increasing intensity of red depicts areas where transmission suitability is marginal (1–3 months), moderate (4–6 months), seasonal (7–9 months) and endemic (10–12 months). For (**d**–**f**), colour-bar scales are similarly grouped, with increasing intensity of red depicting increases in transmission suitability of 1–3 months, 4–6 months, 7–9 months and 10–12 months, and increasing intensity of blue depicting the equivalent decreases. Only areas where differences are statistically significant at the 95% confidence level are plotted.

**Table 1 ijerph-19-13656-t001:** Thermal limits for the four disease vectors considered in this paper, compiled from Ryan et al. [7,8,16].

System	Minimum Temperature (°C)	Maximum Temperature (°C)	Method	Source
*Plasmodium*|*Anopheles*	22.9	27.8	Upper quantile (25%) of the curve R_0_.	[8]
ZIKV|*A. aegypti*	23.9	34	Thermal boundaries for which R_0_ > 0 with a posterior probability >0.975	[7]
DENV|*A. aegypti*	19.9	29.4	Thermal boundaries for which R_0_ > 0 with a posterior probability >0.975	[16]
DENV|*A. albopictus*	21.3	34	Thermal boundaries for which R_0_ > 0 with a posterior probability >0.975	[16]

## Data Availability

Publicly available datasets were analyzed in this study. These data can be found here: Berkeley Earth Monthly Average Temperature data available at http://berkeleyearth.org/data/ (accessed on 11 June 2021). CMIP6: The historical and future climate model data used in this study come from the publicly available CMIP6 datasets that can be accessed via https://data.ceda.ac.uk/badc/cmip6/data/CMIP6 (accessed on 6 September 2021). For all cases, monthly average surface temperature data (variable name: tas) were used. NDVI data: MODIS NDVI data were downloaded from the NASA Earth Observations interface available at https://neo.gsfc.nasa.gov/ (accessed on 31 August 2021).

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
