# Peer review of "Predicting Transmission Suitability of Mosquito-Borne Diseases under Climate Change to Underpin Decision Making"

_ijerph, 2022, doi:10.3390/ijerph192013656_

Round 1
Reviewer 1 Report
The study predicts suitability under climate change for three mosquito-borne diseases: malaria, dengue and Zika virus. The paper is well-written, and of great relevance for public health, entomology and climate change audiences. I have a few comments on the study for the authors to consider.
Introduction
Line 45: remove ‘The’ at the start of the sentence
Line 67-68: It is not clear what the authors mean by ‘there is no research to date that uses the most up-to-date climate projections’. There are plenty of papers that have published using recent climate projections, but it might not necessarily be CMIP6 data. Please can the authors clarify exactly what is meant here
Results
All figures: it is very difficult to see the difference between the shades of red for the two highest categories and the shades of blue for the lowest categories. It would be very helpful if the graphs could have very simple labels on them (e.g. ‘Present day’, ‘SSP1-2.6’, ‘SSP5-8.5’). The maps are very small, it is very difficult to see any of the detail, even when zoomed in; if possible, I suggest that they are printed larger in the paper.
All diseases combined section: it’s not clear to me why the authors have included a combined section in the paper – the mosquito species for malaria are different for dengue and Zika, I am not sure that it makes sense to ‘lump’ them together. Please can the authors provide some (biologically meaning) justification for doing so?
Discussion
Lines 459-461: The authors state that ‘our results reinforce the importance of following the best-case climate change scenario’. Because the maps are very difficult to interpret as they currently stand, it would be useful for the authors to clarify the differences in the SSP1-2.6 and SSP5-8.5 models for the three diseases in the discussion. I think this would definitely reinforce the importance of keeping temperature increases below 1.5°C
Reviewer 2 Report
General comments
The manuscript entitled "Predicting transmission suitability of mosquito-borne diseases under climate change to underpin decision making" by Sargent and colleagues reports the mechanistic model which efficiently evaluates the global shifts in the transmission suitability due to climate change and global warming for three major mosquito borne disease malaria, Zika virus, dengue fever, which are responsible for high mortality and morbidity all over the world. Additionally, the authors present a unified methodology to measure transmission suitability for multiple diseases. Further authors emphasize on the importance of prior prediction of new transmission suitability regions all over the world for present and in 2050 year scenario, which will be very helpful for decision making stakeholders and agencies to be prepared to design and implement effective strategies for mosquito and mosquito borne diseases control programs.
I have some comments addressing a few major and minor issues for the authors to consider.
Major comments:
One of my primary concerns is the Materials and Methods section. Methodology is the key informatics/attraction for the prospective readers of this manuscript. This section needs major revision, rewriting, sectioning and to be concise for easy understanding and readability.
-
Line 114-119 : In this paragraph authors describe the rationale for opting mechanistic approach compared to statistical approach, although In the introduction section under Line 74-84 authors have already elaborated the same. To avoid text/concept redundancy, I suggest deleting Line 114-119.
-
I suggest making sub-sections, concise description for each section ( possibly 4-5 sentences with relevant references or tables) such as
2.1 Identifying transmissible suitable conditions and areas
2.2 Thermal limits for disease transmission suitability
2.3 Climate pathways under study
2.4 Evaluating transmission suitability
2.5 Statistical analysis
Minor comments:
While the overall quality of the language is fine, there are few edits I would like to suggest for the authors to fix. These are listed below :
Line 127-128: I suggest rewriting this sentence, for example “Thermal limits of mosquitoes that spread malaria parasites, Plasmodium (Anopheles gambiae and Anopheles stephensi), Zika virus (Aedes aegypti) and dengue fever (Aedes aegypti and Aedes albopictus) were selected….” . This brings consistency in sentences regarding parasites or viruses ( in bracket mosquito sp spreads it), also if you are mentioning the full scientific name for one mosquito spp. Then it makes sense to maintain the same pattern throughout the sentence.
I suggest maintaining consistency in citing figures throughout text. As per journal instructions , I suggest citing any figure in text as for example Line 225 : “(Figure 1, Scheme a-b, d)” instead of “ the northern temperate zone (Figures 1a, b and d)...” . Similarly, Line 246, 247, 250, 252, 254, 255, 264, 281, 285, 293,316, 320, 326,330,331 needs editing for citing figures.
Line 313 : I suggest maybe “3.4. All diseases combined” could be rewritten as “3.4 Unified disease Transmission suitability”
The discussion and conclusion section can be improved by concising its text length.
Line 495-496 : I suggest rewriting this sentence. Suggest changing phrase “ equal weighting” to “Equally Weighted Index for all three mosquito-borne diseases”
